

# Phosphate solubilizing microorganisms isolated from medicinal plants improve growth of mint

Muhammad Rizwan Tariq[1], Fouzia Shaheen[2], Sharmeen Mustafa[2], Sajid ALI[3], Ammara Fatima[4], Muhammad Shafiq[5], Waseem Safdar[6], Muhammad Naveed Sheas[7], Amna Hameed[7] and Muhammad Adnan Nasir[8]

[1] Department of Food Sciences, University of the Punjab, Lahore, Punjab, Pakistan
[2] Faisalabad Medical University, Faisalabad, Punjab, Pakistan
[3] Department of Agronomy, University of the Punjab, Lahore, Punjab, Pakistan
[4] Lahore College for Women University, Lahore, Lahore, Punjab, Pakistan
[5] Department of Horticulture, University of the Punjab, Lahore, Punjab, Pakistan
[6] Department of Biological Sciences, National University of Medical Sciences, Rawalpindi, Punjab, Pakistan
[7] Department of Diet and Nutritional Sciences, Ibadat International University, Islamabad, Federal, Pakistan
[8] Department of Diet and Nutritional Sciecnes, University of Lahore, Gujrat, Punjab, Pakistan

Corresponding authors
Sajid ALI, sajid.agronomy@pu.edu.pk
Waseem Safdar,
waseem.safdar@yahoo.com

## ABSTRACT

The current research project involves isolation and characterization of PSM (phosphate solubilizing microorganisms) from the rhizospheric soil of certain medicinal plants and to determine their effect on plant growth. Medicinal plants, *Aloe vera, Bauhinia variegata, Cannabis sativa, Lantana camara* and *Mentha viridis* were selected for the isolation of PSMs. Soil status of the selected medicinal plants was also checked. Phosphate solubilizing bacteria (PSB) were observed under stereomicroscope for their morphological characteristics and Gram's staining. Phosphate solubilizing fungi (PSF) were also identified microscopically. Colony diameter, halo zone diameter and solubilization index were determined on PVK agar plates. TLC results indicated that citric acid was the most common acid produced by PSM strains. All strains were found to be non-pathogenic in pathogenicity test. A positive plant growth response to PSM inoculation was observed in all studies. In study 1, individual inoculation of PSM showed a significant increased effect on plant growth parameter *i.e.*, fresh and dry weight, plant height and root and shoot length as compared to control. In study2, composite inoculation of PSM along with different P sources revealed that rock phosphate (RP) with PSM increased growth of plants significantly. The present study suggests that PSM inoculation along with RP amendment can be used as biofertilizer.

## INTRODUCTION

Phosphorous is one of the most essential primary elements (*Glaser & Lehr, 2019*) for plant growth in addition to nitrogen and potassium (*Sharma et al., 2007*). It is a fundamental structural component of plant cell like DNA, RNA and ATP, *etc.* and serves as catalyst in several biochemical processes occurring in plants. Phosphorus nutrition is also
associated with resistance development against diseases in plants (*Bargaz et al., 2018*; *Kaur, Selvakumar & Ganeshamurthy, 2019*; *Bakhtiari, 2014*). Phosphorous is present in soil in both organic and inorganic forms; its content varies in soil from 0.02–0.5% with an average of 0.05% (*Ahmed et al., 2019*).

Phosphorous (P) is the least mobile nutrient in the soil and plants when compared with other macronutrients. It is uptaken by the plants from the soil in the form of phosphate $(PO_4)^{-2}$. Phosphorous is essential for plant growth as it triggers the growth of young plants, accelerate a vigorous start and hastens the maturity. Insufficient supply of P reduced the plant growth. Phosphorous nutrition in plants is linked with some particular growth factors as it strengthens the stalk and stem straw, boosts the formation of the flowers and production of fruits, enhances development of the roots and also has the consequential role in seed formation. P fertilization may enhance the quality of vegetables, fruits, and grain of cereal crops and helps the plant to cope up with adverse environment (*Mawo, Mohammed & Garko, 2016*).

Phosphorus is acquired by microorganisms such as mycorrhizal fungi and PSMs (*Fankem et al., 2006*). Phosphorus solubilizing microorganisms (PSM) have been found in the rhizospheric soil since 1903 (*Khan, Zaidi & Wani, 2007*). In the last few decades, soil-plant-microbe interaction has attracted much attention. Many species of microorganisms have been discovered in soil, particularly in the rhizosphere, and they are recognised to play an essential part in the growth and development of plants. Bacterial species are more efficient than that of fungi in the solubilization of phosphorus(*Anand, Kumari & Mallick, 2016*). Phosphorus solubilizing bacteria (PSB) make 1–50 percent of the total microbial population in soil, while phosphorus solubilizing fungi (PSF) make just 0.1–0.5 percent (*Chen et al., 2006*). Efficiency of bacteria for phosphorus solubilization is comparatively higher than that of fungi (*Anand, Kumari & Mallick, 2016*).

Ectorhizospheric strains from Bacilli and Pseudomonas, as well as endosymbiotic rhizobium strains, have been studied as effective phosphate solubilizers in bacterial communities in the soil (*Kumar, 2016*). *Megaterium, circulans, subtilis, polymyxa, sircalmous, striata and Enterobacter* are the most significant strains of Bacillus (*Kar et al., 2020*; *Richardson & Simpson, 2011*). Nematofungus (*Arthrobotrys oligospora*) can also dissolve phosphate rocks by the *Goswami et al. (2019)*.

Numerous soil bacteria in Pakistan, notably those from Pseudomonas and Bacillus genera, as well as fungus belonging to Penicillium and Aspergillus genera, have the capacity to secrete acids that can transform phosphates from insoluble to soluble forms (*Afzal et al., 2005*). The biochemical properties of some of the most effective phosphate solubilizing bacteria and various phosphate solubilizing fungi isolated from maize rhizosphere showed conversion of insoluble P to soluble P by bacteria were more efficient than that of fungi (*Manzoor, Abbasi & Sultan, 2017*). There were very few studies on phosphate solubilizing bacteria from medicinal plants. *Gupta, Bhatt & Chaturvedi (2018)* studied four PSB on a medicinal plant Aloe barbadensis to enhance aloin-A content and yield. They reported that treatment with PSB significantly increase soil available P, which ultimately resulted in increase in aloin-A yield due to higher plant biomass. In another study, it was reported that a combined use of phosphate solubilising bacteria with TCP can effectively enhance yield

of menthol and essential oil by increasing growth of *Mentha arvensis* (*Prakash & Arora, 2019*). Similar results were obtained on *Stevia rebaudiana* (*Arora et al., 2016*), *Ocimum basilicum* (*Singh et al., 2013*), *Aloe vera* (*Silva et al., 2020*), Asclepias sinaica (*Fouda et al., 2015*), Manihot esculenta (*Ansary et al., 2018*), Teucrium polium (*Hassan, 2017*), Withania somnifera (*Singh & Gaur, 2016*).

The accessibility of soluble phosphate is increased by phosphate solubilizing microorganisms (PSM), and it also enhanced biological nitrogen fixation that increased the growth of plants (*Ponmurugan & Gopi, 2006a*). Trials with phosphate solubilizing bacteria also showed rises in rice yield (*Ordookhani, Sharafzadeh & Zare, 2011*), maize (*Kumar, Bhargava & Ch, 2010*) and other cereals (*Afzal et al., 2005*). *Pseudomonas* spp. increased the quantity of nodules, grain yield, yield components, dry weight of nodules, nutrient uptake, and availability in soybean (*Son, Diep & Giang, 2006*). Seedling length of *Cicer arietinum* was enhanced by phosphate solubilizing bacteria (*Sharma et al., 2007*), though co-inoculation of PGPR and PSM decreased application of phosphate by 50% without altering corn produced (*Yazdani et al., 2009*). Sugarcane yield increased by 12.6% with PSB inoculation (*AYE, PINJAI & TAWORNPRUEK, 2021*). Solitary bacterial application increased the yield biologically, whereas the utilization of mycorrhizae along with same bacteria attained maximum grain weight (*Mehrvarz, Chaichi & Alikhani, 2008*).

In the present era, farmers' views had been changed due to deleterious effect of chemical fertilizers on the environment and also due to increase in knowledge about a good plant soil relationship by using microbial inoculants with proven agronomic effects (*Gupta, Bhatt & Chaturvedi, 2018*). Simultaneously, high cost of phosphate fertilizers makes farmers undercapitalized onerous. Selection of medicinal plant mostly depends on the secondary metabolites they produce and the composition of the root exudates. Studies have shown that endophytic microbiota also contribute towards production of bioactive compounds in medicinal plants (*Tan et al., 2018*; *Kaaniche et al., 2019*). Therefore, diversity of microbial community also play certain role according to soil type, nutritional needs of medicinal plants, and the environment in which they found. *Jensen et al. (2011)* have reported horizontal gene transfer from microorganism to metabolic synthesis pathway of medicinal plants which might affect heterogeneity of the phytochemical profile. Thus, a lot of research is needed to explore biotechnological potential of PSB in medicinal plants to increase P availability, and reduce fertilizers use. The present study was performed to isolate, enumerate and characterise PSM from the selected medicinal plants *i.e., Aloe vera, Bauhinia variegate, Cannabis sativa, Lantana camara* and *Mentha viridis* and to study P solubilization from rock phosphate, single super phosphate and tricalcium phosphate by PSM mint isolates and their effect on growth of mint.

## MATERIALS AND METHODS

### Sample collection

Soil sample were collected from the rhizosphere of five different types of medicinal plants (*Aloe vera, Bauhinia variegata, Lantana camara, Cannabis sativa* and *Mentha viridis*) from the fields of district Lahore, Pakistan. All soil samples were collected in sealed sterilized

plastic envelop and kept at 4 °C till further use. Physico-chemical properties (soil moisture, soil pH, soil texture,) were analysed for all collected soils according to the methodology of *Silva et al. (2020)*.

## Isolation and enumeration of phosphate solubilizing microorganisms

Spread plate method and serial dilution was carried out to isolate PSM from each sample (*Manzoor, Abbasi & Sultan, 2017*). Dispersion was prepared by using 1 g of soil sample in nine mL of autoclaved distilled water. Solution was properly mixed. one mL was taken from already mixed sample and nine mL of sterile distilled water added in it to prepare $10^{-2}$ dilution. Other dilutions $10^{-3}$, $10^{-4}$, $10^{-5}$, $10^{-6}$, $10^{-7}$ and $10^{-8}$ serials were prepared for every sample. For fungal growth, 0.1 mL was taken from every dilution and spread on potato dextrose agar (PDA) slants for fungal growth and then transferred to Pikovskaya's agar medium (PVK) having insoluble Tricalcium phosphate (5 g/L). For bacterial growth, 0.1 mL from each sample was spread on PVK media (M520-VWR with 1.5% agar gel, pH was adjusted to 7.0 before autoclave). Incubation was done for seven days at 27–30 °C. The purified colonies were maintained in glycerol stocks. pH was continuously monitored throughout the incubation. Colonies showing phosphate solubilizing zones (due to solubilization of Tricalcium phosphate) were transferred to fresh Pikovskaya's agar medium (PVK) for morphological characterization.

Phosphate solubilization index (SI) indicating bacterial ability to solubilize phosphate was calculated by using the following formula;

$$Phosphate\ Solubilization\ Index\ (SI) = \frac{Total\ Diameter\ (Colony + halozone)}{Colony\ diameter}.$$

## Physical and chemical characterization of PSM

Colonies of PSB were studied for five various morphological attributes (margin of colony, colour, surface texture, elevation surface form) under stereomicroscope. PSB were identified by Gram's staining. Phosphate solubilizing fungi were grown on Czapek medium agar plates. Colour of colonies developed on media was noted. TLC was performed for organic acids determination. For pathogenicity test, tobacco plant was chosen to get clear and early response of hypersensitivity symptoms.

## Inoculation effect of PSM on mint growth

The effect was studied in two different experiments. The first experiment was unifactorial, completely randomized and individual bacterial and fungal isolates; obtained by harvesting microorganisms at their exponential phase (at concentration $10^7$–$10^9$) by centrifugation and dissolved in phosphate saline buffer; were inoculated in treatments. Uninoculated treatment served as control. Three replications were used for each treatment. The second experiment was bi-factorial completely randomized with three replicates. Cumulative behavior of selected fungal and bacterial strains on mint plant growth and their interaction with different phosphate sources *i.e.,* rock phosphate (RP, Merck, Germany), single super phosphate (SSP, Safi Group, Pakistan), and tricalcium phosphate (TCP, Sigma, Germany) was studied. RP and TCP were employed at level of 2 gm/kg and 50 mg of SSP were added to

the respective treatments. Uninoculated and non-P fertilized treatment served as control. Fresh weight of shoots, fresh weight of roots, plant height, root length, dry weight of shoots and roots were taken after 24 h and 72 h of plant growing in pots, till the weight of each dried sample became constant.

## Statistical analysis

The results were analysed by using analysis of variance (ANOVA). The means were depicted graphically using the Statistical Package for Social Sciences (IBM-SPSS ver. 25.0; IBM, Armonk, NY, USA).

## RESULTS

### Physico-chemical analysis

Physico-chemical analysis (soil moisture and soil pH) of the selected five medicinal plants *i.e., Aloe vera, Bauhinia variegata, Cannabis sativa, Lantana camara* and *Mentha viridis* (five rhizospheric samples of each plant) are shown in the Table 1. Thirty-seven PSM strains were isolated from rhizosphere of different medicinal plants. Fungi and PSB represented 38.2% and 4.14% among entire microbial population correspondingly. The highest number of phosphate solubilizing bacteria ($301.3 \pm 1.5$) was found to be associated with *Mentha viridis* and *Bauhinia variegata* ($125.3 \pm 1.6$) while the highest phosphate solubilizing fungi (55.3%) were found to be associated with *Aloe vera.* The least number of phosphate solubilizing fungi was associated with *Bauhinia variegata* and *Cannabis sativa* while least phosphate-solubilizing bacteria was associated with *Aloe vera* (Table 2).

### Morphological characterization

Total 22 PSB were separated from the rhizosphere of plants *i.e., Bauhinia variegata, Aloe vera, Cannabis sativa, Lantana camara* and *Mentha viridis*. PSB colonies were studied for their morphological features *e.g.,* elevation, colour, surface form, surface texture and margins. Bacterial strains and their shapes were identified by gram staining. PSM showed gram variability. Bacteria were either cocci or bacilli, majority were gram negative and some were gram positive (Fig. 1). A total of 15 PSF were isolated. Fungal colonies identification through morphology and microscopic analysis showed that most PSF genera belong to *Penicillium* and *Aspergillus*.

### Chemical characterization

Colony phosphate solubilizing zone diameter and solubilization index of the isolated PSB and PSF from the five medicinal plants were measured during seven days of incubation (Fig. 2). It was observed that there was a steady increase in the colony and phosphate solubilizing zone diameter during the incubation period (Fig. 3). Fluctuations were observed in solubilization index. Results showed that CB4 having solubilization index of 2.9 was highly effective bacterial phosphate solubilizer on Pikovskaya's agar plates. Among fungi *Penicillium sp.* showed best results with 2.23 solubilizing index. PSM isolates were studied for their characteristic of lowering the pH of broth medium. Present results showed that most PSM reduced pH of Pikovskaya's broth agar with respect to the control.

**Table 1  Physico-chemical properties of soil samples from five medicinal plants.**

| S. No. | Medicinal plant | Soil moisture | Soil pH |
|---|---|---|---|
| 1 | *Aloe vera* | 0.914 | 7.84 |
| 2 | *Bauhinia variegata* | 2.41 | 7.69 |
| 3 | *Cannabis sativa* | 2.70 | 7.98 |
| 4 | *Lantana camara* | 1.03 | 7.42 |
| 5 | *Mentha viridis* | 1.38 | 7.23 |

**Table 2  Population of total bacteria, fungi, total phosphate solubilizing bacteria (PSB) and phosphate solubilizing fungi (PSF) of selected rhizosphere samples of medicinal plants.**

| Sample | Total bacteria | Total fungi | TPSB | %PSB | TPSF | %PSF |
|---|---|---|---|---|---|---|
| *Aloe vera* | $60.7 \pm 2.7$ | $2.53 \pm 0.7$ | $0.8 \pm 0.3$ | 1.31 | $1.4 \pm 0.27$ | 55.3 |
| *Bauhinia variegata* | $125.3 \pm 1.6$ | $1.2 \pm 0.37$ | $1.73 \pm 0.3$ | 1.38 | $0.26 \pm 0.14$ | 20.63 |
| *Cannabis sativa* | $63.5 \pm 2.4$ | $3.53 \pm 0.8$ | $10.73 \pm 3.1$ | 16.89 | $0.53 \pm 0.15$ | 15.01 |
| *Lantana cammara* | $301.3 \pm 1.5$ | $3 \pm 1.1$ | $9.13 \pm 2.2$ | 3.03 | $1.66 \pm 0.72$ | 55.3 |
| *Mentha viridis* | $66.6 \pm 1.6$ | $3.4 \pm 1.2$ | $3.2 \pm 0.6$ | 4.80 | $1.4 \pm 0.33$ | 41.17 |

Notes.
Each value is the mean of 15 replicates TPSB, total phosphate solubilizing bacteria TPSF, total phosphate solubilizing fungi % PSB, percent phosphate solubilizing bacteria % PSF, percent phosphate solubilizing fungi.

Fluctuations in pH drop were also noted during seven days. Some PSM showing greater phosphate solubilizing zones that did not reduce the pH substantially. Lowest pH recorded for PSB is 3.0 by CB4 while lowest pH for PSF is 3.1 by *A. versicolar*. Results indicated that bacteria and fungi were likewise effective in reducing pH (Fig. 4).

The most efficient PSB and PSF strains of the selected medicinal plants were checked for the organic acid production by thin layer chromatography. Organic acids production was determined by calculating the Retardation factor ($R_f$) value between ($0.14-0.37$). Twelve most efficient fungal strains and thirteen bacterial were discovered to be non-pathogenic in pathogenicity analysis.

## Inoculation effect of PSM on Mint plant

In the first case, inoculation effect of PSM on plant depicted that height of plants were affected significantly ($P \leq 0.05$) with PSM inoculation. Highest value for height of Mint plant was obtained with 1MB (20 cm). All treatments of PSM were significantly ($P \leq 0.01$) different from control for root length. Highest root length was observed with *Aspergillus versicolor* (MF) (15 cm). A highly significant ($P \leq 0.01$) effect on fresh and dry weight of mint in treatments were observed. 2 MB (1.55 g) showed significantly highest fresh weight and highest dry ($P \leq 0.01$) as compared to control. In the second case, a significant increase ($P \leq 0.01$) in plant height of mint with PSM inoculation and treatments (control, RP, TCP, SSP) was observed. The highest results were obtained with RP + PSM (24.3 cm) and lowest for SSP (15.2 cm) without PSM as compared to control (10.2 cm). Result also showed a significant increase ($P \leq 0.01$) in root length of mint with PSM inoculation and treatments as compared to control. Highest results were obtained with RP + PSM (13

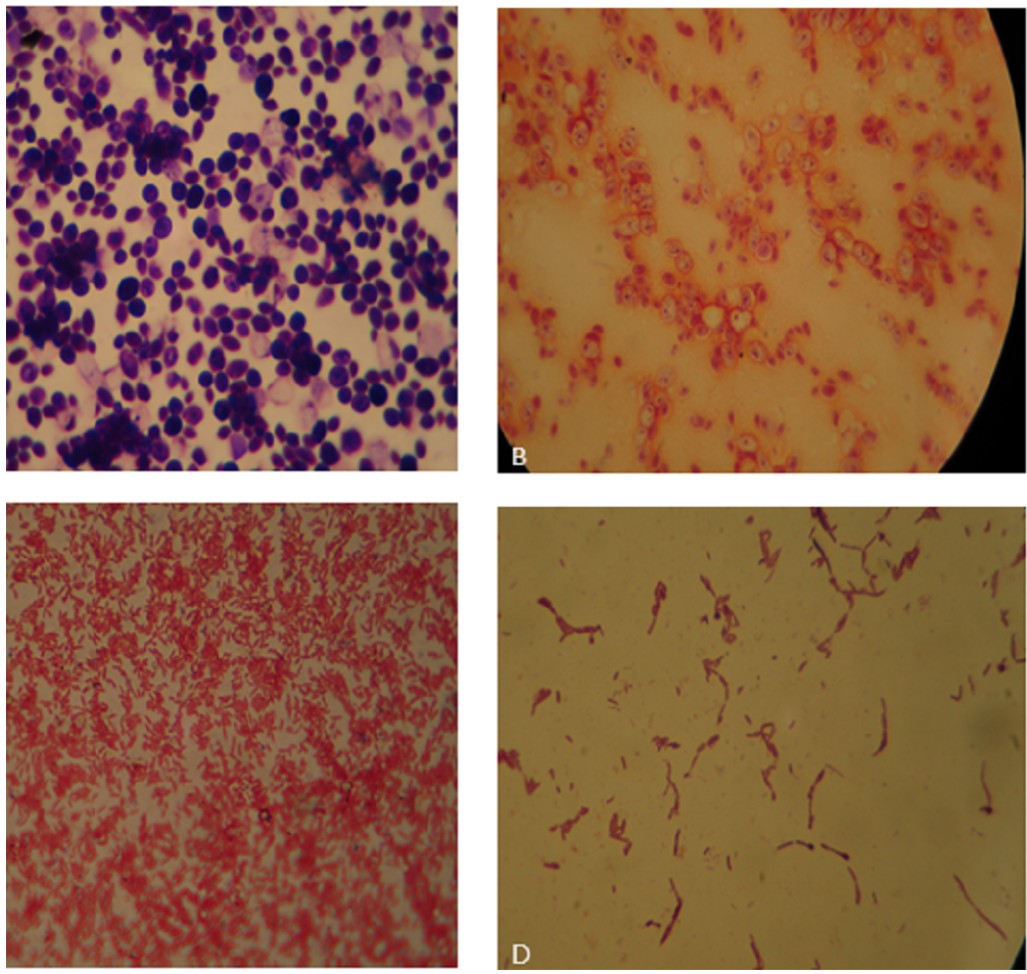

**Figure 1** (A) *Bauhinia* bacteria BB1showing Gram positive cocci; (B) *Lantana* bacteria (LB2) showing Gram negative cocci; (C) *Cannabis* bacteria (CB1) showing Gram negative rods; (D) *Aloe vera* bacteria (AB) showing Gram positive rods.

cm) and lowest for TCP (three cm) without PSM as compared to control. Results also showed highly significant effect of PSM on fresh and dry weight (g) of mint as compared to uninoculated ones.

## DISCUSSION

The soil selected from the rhizosphere of medicinal plants (MPs) *i.e., A. vera, B. verigata, C. sativa, L. camara* and *M. viridis* showed variation in chemical and physical nature to some extent from soil to soil. To some extent variation in soil pH and moisture were found during study. These soil properties indicated that all the soil samples were alkaline with pH range of 6.9–8.2. Variation of soil texture was also found from clay loam, loam to sandy loam. *Qureshi et al. (2001)* also showed that texture of soil varied from clay loam (43%) to sandy loam (10%) and loam (47%) with pH ranges from 6.9–8.5. Iron and aluminium reaction with phosphorous in low pH or acidity results in reduction of soil phosphorus while in
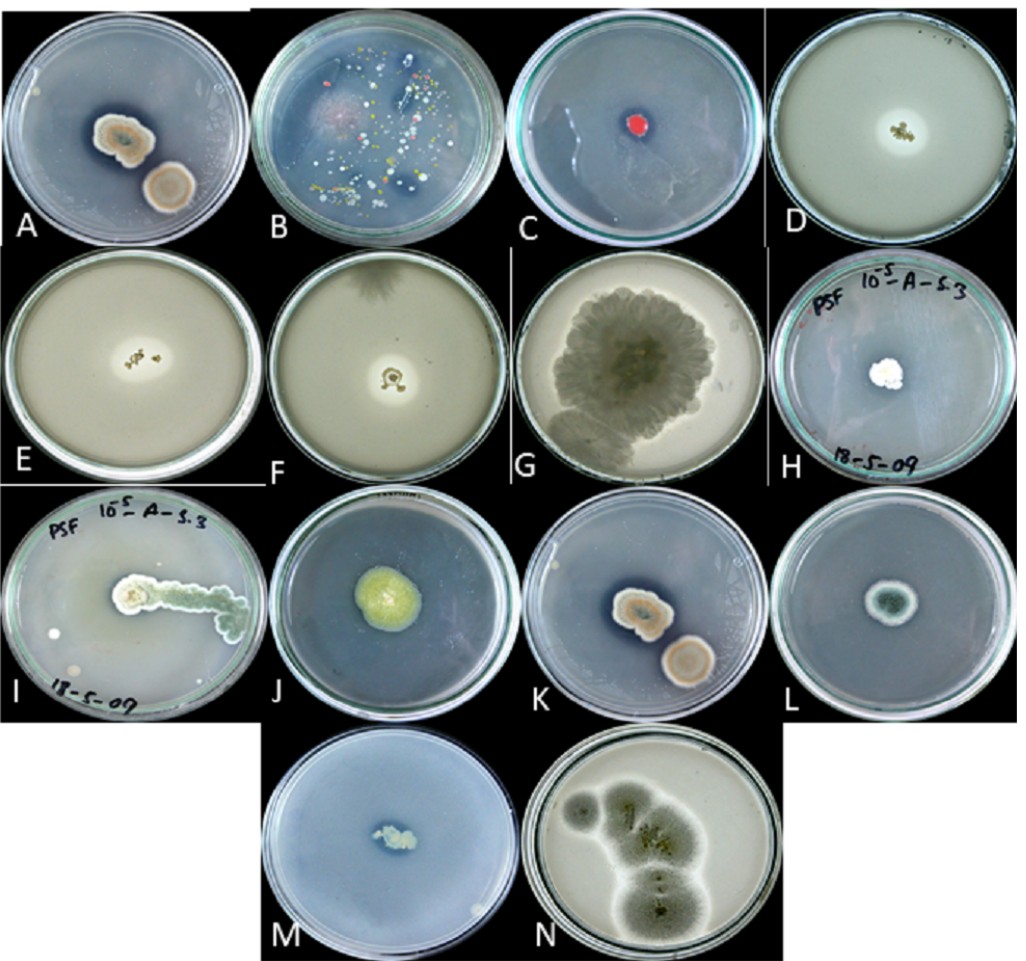

**Figure 2** P solubalized and halozone produced in mixed culture by (A) fungi and (B) bacteria; P solubilized and halozone produced by (C) B2 and (D) B3; P solubilized and halozone produced by (E) B4 and (F) B6; P solubilized and holozone produced by (G) *Fusarium sp.* and (H) *Penicillium sp.*; P solubilized and halozone produced by (I) *Penicillium sp.* and (J) *Penicillium sp;*. P solubilized and holozone produced by (K) *Aspergillus versicolor* and (L) *Penicillium sp.*; P solubilized and halozone produced by (M) MB and (N) *Aspergillius terreus.*

alkaline or high pH soils, tricalcium phosphate $(Ca_3PO_4)_3$ is formed which ultimately reduce the soil phosphorous availability to plants (*Islam et al., 2019*; *Adeleke, Nwangburuka & Oboirien, 2017*; *Khan et al., 2010*). This process is called as phosphorous fixation. So, alkalinity of soil suggests the unavailability of phosphorous to the plants.

Enumeration studies showed PSM variation associated with the rhizosphere of 25 samples of five different medicinal plants. PSF colonies ranged from 15.01% (*C. sativa*) to 55.3% (*A. vera* and *L. camara*) out of the total fungal colonies, while that of PSB varied from 1.31% (*A. vera*) to 16.89% (*C. sativa*) out of the total bacterial colonies. The rhizosphere of maize from four Quebec soils constituted 26–46% of microbial population (*Antoun, 2012*) while *Sembiring & Sabrina (2022)* stated that PSB and PSF constituted 7.1–55.6% and 8.1–57.9% respectively. In the north part of the Iran, the PSB count was 3.98% among
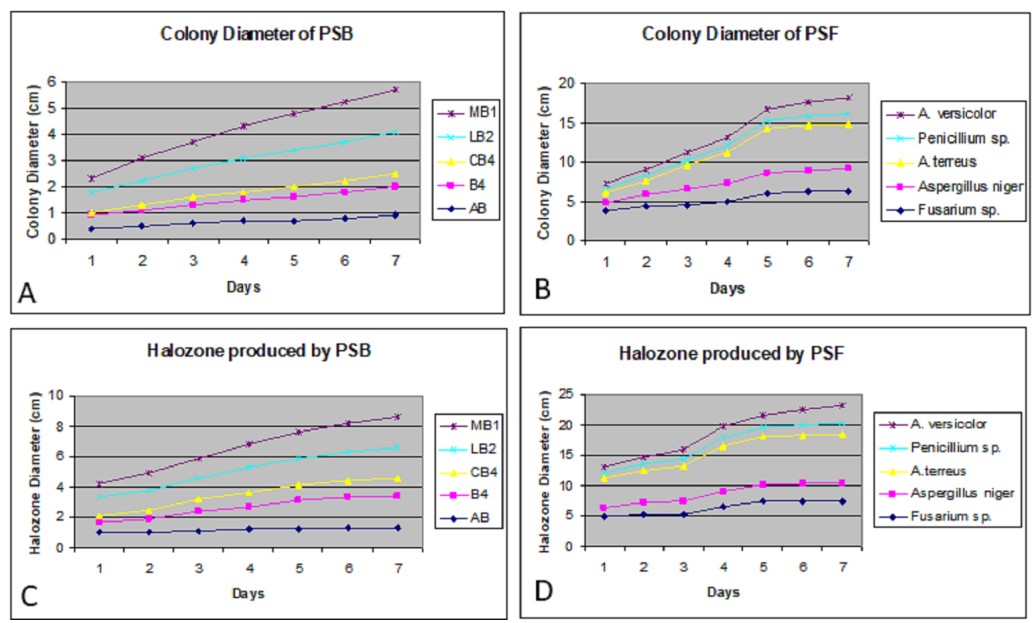

**Figure 3** Colony diameter of phosphate solubilizing bacteria (A) and fungi (B) on Pikovskaya (PVK) agar plates during seven days of incubation. Halozone produced by phosphate solubilizing bacteria (C) and fungi (D) on Pikovskaya's (PVK) agar plates during seven days of incubation.

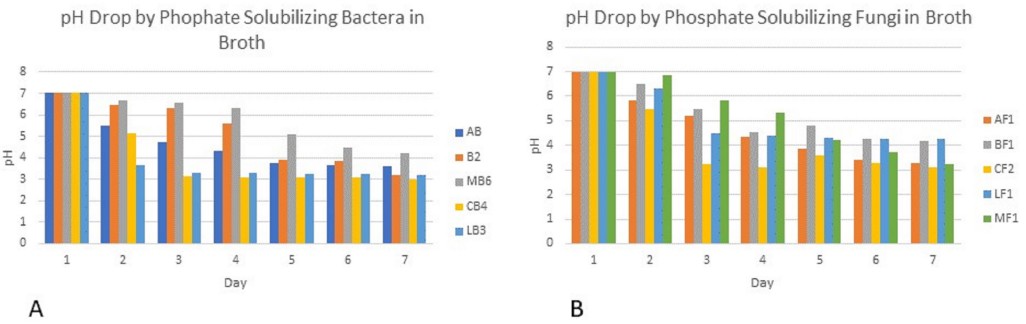

**Figure 4** PH of Pikovskaya's broth medium by phosphate solubilizing (A) bacteria and (B) fungi.

total bacteria (*Fallah, 2006*). This difference may be due to different soil conditions of their studies. As varieties of plant species have differential capabilities to exude chemicals, they regulate population of rhizosphere (*El Maaloum et al., 2020*). The variation of microbial distribution in the rhizospheric soil of MPs is similar to the study of *Brady (1990)* who concluded that physical and chemical properties of soil influence the distribution of the microbial population in the soil. Similar results were shown by *Baby et al. (2001)* that the variation in population level of phosphobacteria in the soil rhizosphere might be due to many soil factors. PSB showed different morphological characters and Gram's staining of Pikovskaya's agar medium. The majority were gram negative rods but gram positive rods and gram negative and positive cocci were also found (*Baudoin, Benizri & Guckert,*

*2002*; *Kumar, Bhargava & Ch, 2010*; *Joy et al., 2018*). Fungi associated with phosphate solubilization were also identified. The results showed that *Aspergillus* and *Penicillium* are the most common genera. Similar results were shown by *Yin et al. (2017)*.

When agar plates were used for studying PSMs, it was found that microorganisms used for solubilization of phosphates formed pure zone by the solubilization of floating tricalcium phosphate. Measurements ranged from 1.2–2.3 cm for bacteria and 1.5–3.7 cm for fungi. The solubilization index varied from 1.17–2.90 cm for bacteria and 1.06–2.23 cm for fungi. Similar results were shown by *Edi-Premono, Moawad & Vlek (1996)* and *Kumar, Bhargava & Ch (2010)*. Some of the PSM strains lost their ability to form holozone on Pikovskaya's agar medium on sub culturing. This was also observed by *Richardson & Simpson (2011)*, *Alam (2001)*, and *Walpola & Yoon (2013)*. Phosphate solubilizing fungi formed larger holozones as compared to bacteria. Similar results were observed by *Alam (2001)*, *Chabot, Antoun & Cescas (1996)* and *Sembiring & Sabrina (2022)*.

Broth pH studies showed a tremendous decrease in pH from 7.0 to 3.0 by Cannabis bacterial strain CB4 and 3.1 in the case of Cannabis fungal strain. Similar results were also observed in the experimental data of *Alam (2001)*, *Bar-Yosef et al. (1999)* and *Othman & Panhwar (2014)*, which showed that continuous growth of these isolates for seven days results in drop in pH of the medium. This drop in pH indicates the production of the acids that have lowered the pH and change the alkalinity of the broth to acidity. Similar results were shown by *Walpola & Yoon (2013)*. Fluctuations in pH drop were also noted during seven days. Some PSM showing larger holozones did not lower the pH significantly. *Flatian, Anas & Sutandi (2021)* described that proton excretion with ammonium ion assimilation is the most effective description for microbial solubilization under low acid production.

Organic acid production was detected by thin layer chromatography. The purpose of the TLC was to detect the organic acids production using the most efficient PSB and PSF strains identified from the rhizosphere of MPs. The results showed that citric acid was the major organic acid produced by PSMs in this study. The results of *Audipudi, Kumar & Sudhir (2012)* revealed that the most common organic acid yielded by different isolates were isovaleric, succinic, lactic, acetic and isobutyric and acetic acids. Most bacterial strains produce two or more organic acids, while *A. niger* was limited only to the production of succinic acid. The results were reproduced by *Rawat et al. (2021)*; it was found that phosphobacteria yields organic acids, like monocarboxylic acid (formic and acetic acid), hydroxy monocarboxylic (lactic, glucene, and glycolic acids), ketoglucenic, monocarboxylic, decarboxylic (oxalic), tricarboxylic, hydroxy (maleic and malic acids) and hydroxy (citric acid) dicarboxylic acids for the solubilizing inorganic phosphate compounds. PSM studies have indicated that they have the power to affect plant growth positively. Inoculation effect of PSM has been found to impose a positive impact on growth of plants. *Panhwar et al. (2014)* reported significant increase in the grain and straw yields as a consequence of grafting. The authors also reported improvement in phosphate absorption in comparison with that of control by phosphate solubilizing microorganisms both with and without chemical fertilizers. In Study 1, plant heights of all treatments increased with PSM inoculation compared to positive controls. Inoculation of PSM (14.5–20.0 cm) had a greater effect on plant height, in contrast 2 MB proved to be the promising strain. *Joy et*

*al. (2018)* reported a significant increase in sorghum plant height by inoculating various bacterial strains confirmed by *Alagawadi (1996)*.

*Pande et al. (2017)* found 105 MB to be the most effective cultivar for increasing plant height but contrarily *Wang et al. (2015)* found *A. niger* to be the most effective PSM for increasing plant height. Phosphate solubilizing bacteria increased the length of Cicer arietinum seedlings (*Sharma et al., 2007*).

Based on the results of *Mehrvarz, Chaichi & Alikhani (2008)*, it can be concluded that if bacteria are applied in the absence of inorganic phosphate fertilizer, it may be adequately effective for increasing biomass production to a satisfactory level, and hence can be contemplated as an appropriate substitute for inorganic phosphatic fertilizer in the organic systems of agriculture farming.

A greater effect of SCM inoculation in the roots was noticed. The longest root length was measured with the MF (15 cm). P uptake is not only involved in promoting growth, but PSM also produces growth promoting substances such as gibberellins, which affect plant root growth. Growth promoting effects of bacteria may include the production of phytohormones (*Chabot, Antoun & Cescas, 1996*). *Ke et al. (2019)*. A noticeable effect on freshly harvested mint mass was observed with all treatment options (Study 1) and maximum fresh weight was obtained with 2 MB. *Ponmurugan & Gopi (2006b)* claimed that phosphobacteria enhance plant growth by biosynthesizing plant growth substances. All strains of phosphobacteria were able to dissolve inorganic phosphate. Bacteria solubilizing phosphate are capable of producing physiologically active auxins, which can have a pronounced effect on plant growth (*Mukhopadhyay, Thapa & Firoz, 2021*; *Shokri & Emtiazi, 2010*). It was reported that the use of *P. fluorescens* and *B. megaterium* bacteria as seed inoculants would improve seed germination and the growth of seeds and seedlings would increase in yield as well (*Sharma et al., 2007*).

A greater effect on mint dry weight was noticed with all treatment options (Study 1), whereas, the highest dry weight was measured at 2 MB. All inoculated preparations were noticeably different from non-inoculated controls, furthermore, an increase in corn dry matter was found due to Bacillus subtilis (*Chen et al., 2006*). Quite interestingly, *A. niger* was found to be the most effective strain for increasing dry weight in maize (*Arihara & Karasawa, 2000*). It was discovered that Pseudomonas sp. improved dry weight of nodules in soybean crops (*Son, Diep & Giang, 2006*).

Study 2 observed the cumulative effect of PSM, phosphorus sources and their interactions in which it was found that PSM inoculation has a significant effect on the growth of mint plants. All RP, TCP and SSP treatments showed a significant effect on mint height. The highest value of plant height ($14.5 \pm 0.92$ cm) was observed when using RP + PSM compared to control. A pot experiment by *Afzal & Bano (2008)* showed that in all inoculations with P and with double seeding without P, the plant height was significantly higher than with other treatments. A single inoculation with Rhizobium/PSB without P fertilization did not increase plant height compared to control. Thorn length increased with P fertilization, double inoculation and combined application of + P inoculants. These results confirmed previous findings (*Abaid-Ullah et al., 2015*; *Biswas, Ladha & Dazzo, 2000*; *Naeem et al., 2018*), which reported an increase in plant height and the number of spikelets per ear of

various crops during microbial inoculation. PSM inoculation also led to an increase in root length. Various sources of phosphate also affect the length of the mint roots.

The greatest root length was observed at RP + PSM. *Arangarasan, Palaniappan & Chelliah (1998)* and *Ke et al. (2019)* also found longer root lengths with the addition of P sources. PSM inoculation with phosphorus sources also showed a significant effect on fresh mass of mint plants (Study 2). The highest fresh weight was found with RP + PSM (2.7 cm). TCP and SSP with PSM also showed a significant effect on wet weight compared to control. More or less similar results were reflected in studies of *Parmar & Dufresne (2011)* on barley, *Yanni & Dazzo (2010)* in rice, *Chabot, Antoun & Cescas (1996)* in maize. An increase in the dry mass of plants was found upon inoculation with PSM. The highest dry weight was found with RP + PSM. This finding is similar to the results of *Pande et al. (2017)*, *Alam (2001)*, *Sarajuoghi et al. (2012)*, and *Arihara & Karasawa (2000)*. The results of Study 2 showed that RP + PSM has a very significant effect on plant height, root length, wet and dry weight. PSB dissolves phosphate fixed in the soil and applied phosphates, resulting in increased crop yields (*Gull et al., 2004*). Direct PR application is usually ineffective in the short term for most annual crops (*Roberts & Johnston, 2015*) in contrast to acid-producing microorganisms which can enhance the dissolution of phosphates in rocks (*Bargaz et al., 2018*). A study of soil pH at harvest time showed that there were no significant differences between treatments in all cases (*Tao et al., 2008*).

To some extent decrease in pH was due to the PSM inoculation in the sterilized sand pots. This may be attributed to the fact that soil microorganisms other than PSM may produce ammonia which resist change in pH due to organic acids produced by PSM. Similar results were found by *Othman & Panhwar (2014)*.

## CONCLUSION

The findings of the present study suggested that inoculation of native phosphate solubilizing microbes from medicinal plants in rhizosphere can be exploited for improved plant growth. Enhanced growth of plants by PSM can replace commercially used synthetic fertilizers, which are very expensive. These results also suggest that composite PSM inoculation with RP as biofertilizer can significantly enhance growth of plants. Still, more research is needed in this field such as field evaluation and effect of PSM along with mycorrhiza and nitrogen fixing microorganisms on growth of plants may also be studied.

### Funding
The authors received no funding for this project.

### Competing Interests
The authors declare there are no competing interests.

## Author Contributions

- Muhammad Rizwan Tariq conceived and designed the experiments, performed the experiments, prepared figures and/or tables, and approved the final draft.
- Fouzia Shaheen analyzed the data, authored or reviewed drafts of the article, and approved the final draft.
- Sharmeen Mustafa analyzed the data, authored or reviewed drafts of the article, and approved the final draft.
- Sajid Ali performed the experiments, prepared figures and/or tables, and approved the final draft.
- Ammara Fatima analyzed the data, prepared figures and/or tables, and approved the final draft.
- Muhammad Shafiq analyzed the data, authored or reviewed drafts of the article, and approved the final draft.
- Waseem Safdar conceived and designed the experiments, prepared figures and/or tables, and approved the final draft.
- Muhammad Naveed Sheas performed the experiments, authored or reviewed drafts of the article, and approved the final draft.
- Amna Hameed analyzed the data, authored or reviewed drafts of the article, and approved the final draft.
- Muhammad Adnan Nasir performed the experiments, authored or reviewed drafts of the article, and approved the final draft.

## Data Availability

The raw data are available in the Supplemental File.

## Supplemental Information

Supplemental information for this article can be found online at http://dx.doi.org/10.7717/peerj.13782#supplemental-information.

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
