# Peer review of "Phosphate solubilizing microorganisms isolated from medicinal plants improve growth of mint"

_PeerJ, doi:10.7717/peerj.13782_

## Round 0.1 · original submission · Major Revisions

> Please rewrite the lines 44 to 46 to make it more clear.

> The most significant strains of Bacillus include megaterium,
circulans, subtilis, polymyxa, sircalmous, striata (Pseudomonas), and Enterobacter (Kar et al. 2020; Richardson & Simpson, 2011). Phosphate rocks can also be dissolved by the nematofungus Arthrobotrys oligospora (Goswami et al. 2019).

Make these lines clear. from 68 to 70.

> In sample collection section, author should clearly mention the area from where samples were collected.
> In the isolation process, reference is missing and rewrite the sentence from 106 to 108. How authors isolated bacteria and fungi from sample. In my view point, first we isolate fungi on PDA and than PVK medium to verify the P solubility. Please explain the process clearly.

> In the inoculation process, what are the sources of rock phosphate, SSP and TCP. Also mention the company name for chemicals used.

> How author measured the solubilization index?? Mention the formula

> Discussion is poorly written, author should improve the discussion.
> Also incorporate all comments mentioned by reviewers.

Reviewer 1 ·

Basic reporting

Present study was conducted to evaluate the effect for phosphate solubilizing microorganisms from certain local medicinal plants on mint growth. The idea of this research was quite interesting and worth publishing. Results of this study was also quite comprehensive and satisfying, clear and unambiguous. sufficient literature is cited. few suggestions have been made to improve introduction portion. Overall manuscript was well written. However, there is few missing information in methodology, which needs to be clarified. A major revision is suggested to improve this manuscript before accepting for publication in this journal.

Experimental design

experimental design was quite sufficient, up to the mark. research question was well defined and analysed. sufficient details of some experiments have given, however minor details are missing which have been pointed out in general comments.

Validity of the findings

Results were rightly validated, raw data was also quite sufficient to verify the results. conclusions were well drawn in relation to concept of this work.

Additional comments

In introduction section, there is no study cited about phosphate solubilizing bacteria from medicinal plants. Authors need to report few studies on PSM from medicinal plants.
Why you have conducted this piece of work. Write some importance/significance of this study in introduction before narrating objectives.
All Scientific names should be italic.
Line 95: Its batter to write “soil samples of five different types” rather than “Samples of soil of five different types”
Line 96: Sample collection is not clearly stated. Mention time and place clearly from where samples were collected. Also mention sampling strategy, and storage.
Line 104: mention concentration and pH of the medium
Line 105: mention quantity of tricalcium phosphate
Line 131: Statistical software used was very outdated version. Use some latest version of statistical software such as SPSS 25.0 ver. And see if there are difference in analysis.
Line 133: under heading of physiochemical analysis, there is no physical parameters shown. Mention clearly what type of physiochemical analysis was conducted.
Line 136: continue with line 137. Do not use very small paragraphs.
Line 139: explain your results with some numerical data such as percentages ± SD. Similar for other results also..
Line 170: at no place, authors have mention what plant was used for treatment. Write name of the plant.
Line 172: highest value of which plant?
Line 173: What are the treatments, dosage, concentrations, how you applied treatments. Mention clearly. What age of the plant you selected for treatments?. Also clarify, that have you applied the whole soil or just microorganisms. If you have used only microorganisms, how you managed to remove culture media from them?
Line 189: author have mentioned in discussion the quality and texture of soil, while these was not results and methods motioned to analyses it. If you have used some one’s protocol, please cite its reference clearly with explaining modification if made.
Line 282: Write numeric values also
Line 311: please restructure the sentence. It is unclear statement.

Reviewer 2 ·

Basic reporting

no comment

Experimental design

no comment

Validity of the findings

no comment

Additional comments

Abstract:
1. Lines 20-21, please add English names of the medicinal plants too.
Introduction:
1. Lines 37-38, Please re-write the sentence “Phosphorous is one of the most essential elements (Glaser and Lehr, 2019) for plant growth after nitrogen (Sharma et al. 2007)” as “Phosphorous is one of the most essential primary elements (Glaser and Lehr, 2019) for plant growth in addition to nitrogen and potassium (Sharma et al. 2007).
2. Lines 65-67 are not clear, “The most potent P solubilizer strains from the bacterial genera Rhizobium, Bacillus, Enterobacter and Pseudomonas, as well as fungal sp. from the genera Penicillium and Aspergillus (Khan et al. 2010). Rewrite or delete these lines.
3. Lines 73-76, “The biochemical properties of ten of the most effective phosphate solubilizing bacteria and three fungus (PSM) isolated from maize rhizosphere have been investigated. In the conversion of insoluble P to soluble P, bacteria were shown to be more efficient than fungi (Manzoor et al. 2017)” Rewrite as “The biochemical properties of ten of the most effective phosphate solubilizing bacteria and three fungi (PSM) isolated from maize rhizosphere showed that bacteria converted insoluble P to soluble P more efficiently than that of fungi (Manzoor et al. 2017)”.
Materials and Methods:
1. Line 105, mention the concentration of TCP
2. Line 104-105, mention the recipe of Pikovskaya’s agar medium (PVK) used for isolation
3. Lines 117-127, how was the growth of the plants studied, field or pot experiments were done for this purpose?
4. Experiment duration has not been mentioned i.e., how old seedlings were used for the experiments, how many days after sowing were the plants harvested?
5. Lines 120-121, concentration of selected fungal and bacterial inoculum have not been mentioned.
Results:
1. Line 150, the method for identification of fungal clone has not been given, whereas in figure 4 it has been mentioned that fungal clones belong to Penicillium and Aspergillus.
2. Figure 4 on Y axis please replace “change in pH” with “pH”
3. Results of effect on inoculum on plant growth have been mentioned in the results, but no data is available for these. These should be presented either graphically or in the tabulated form.
4. Lines 166-169, the results of organic acid production by the strains have not been clearly mentioned, no Rf values given. The data should be presented in the form of tables or graphs.
5. Please replace “halozone” with “phosphate solubilizing zone” in whole manuscript including figures and tables.
6. Please provide morphological data of all isolated strains in tabulated form.
Discussion:
1. Line 188: pH range of 6.90-8.2. Please stick to same pattern of figures after the decimal point, either 1 or 2.
2. Line 190: clay loam (943 %), looks like typographical mistake, please check and rectify.
3. Line 191: ranges from 9.9 to 8.5. In the line 188 it is “-” and in line 191 it is “to” between the figures of a range. Please stick to the same pattern and check and rectify in the whole manuscript.
4. Line 218: Font color of “Some” needs to be changed to black.

Tables and Figures:
2. Table 2: Please replace the title “Population of total bacteria and fungi and population of total phosphate solubilizing bacteria (PSB) and phosphate solubilizing fungi (PSF) of selected rhizosphere samples of medicinal plants” to “Population of total bacteria, fungi, total phosphate solubilizing bacteria (PSB) and phosphate solubilizing fungi (PSF) of selected rhizosphere samples of medicinal plants”
General Remarks:
1. Replace “ml” with “mL” in whole manuscript.
2. At some places there is space between the value and the unit and it some places this space is missing [e.g., line 172 (20cm) and line 178 (24.3 cm)], please stick to the same style in whole manuscript.
3. In methodology it has been mentioned that ANOVA was used for statistical analysis, whereas tables and figures don’t contain and probability values that show the significance of the means.
4. No lettering has been done and there is no significance test employed to separate the treatment means.

---

## Round 0.2 · Minor Revisions

In introduction section, lines 72 to 74 are incomplete.

Lines 90 to 92, scientific names should be italic. Please check the article for this carefully.

Reference for phosphate solubilization index formula is missing. Moreover, the formula should place in the material and methods section, not in the results.

Reviewer 2 ·

Basic reporting

no comment

Experimental design

no comment

Validity of the findings

no comment

Additional comments

Authors have made all recommended changes accordingly. This manuscript can be accepted for publication as it is.

---

## Round 0.3 · accepted · Accept

All suggestions are incorporated.